# Deconvolutional Paragraph Representation Learning

**Yizhe Zhang**   **Dinghan Shen**   **Guoyin Wang**   **Zhe Gan**   **Ricardo Henao**

**Lawrence Carin**
Department of Electrical & Computer Engineering, Duke University

## Abstract

Learning latent representations from long text sequences is an important first step in many natural language processing applications. Recurrent Neural Networks (RNNs) have become a cornerstone for this challenging task. However, the quality of sentences during RNN-based decoding (reconstruction) decreases with the length of the text. We propose a sequence-to-sequence, purely convolutional and deconvolutional autoencoding framework that is free of the above issue, while also being computationally efficient. The proposed method is simple, easy to implement and can be leveraged as a building block for many applications. We show empirically that compared to RNNs, our framework is better at reconstructing and correcting long paragraphs. Quantitative evaluation on semi-supervised text classification and summarization tasks demonstrate the potential for better utilization of long unlabeled text data.

## 1 Introduction

A central task in natural language processing is to learn representations (features) for sentences or multi-sentence paragraphs. These representations are typically a required first step toward more applied tasks, such as sentiment analysis [1, 2, 3, 4], machine translation [5, 6, 7], dialogue systems [8, 9, 10] and text summarization [11, 12, 13]. An approach for learning sentence representations from data is to leverage an *encoder-decoder* framework [14]. In a standard autoencoding setup, a vector representation is first *encoded* from an embedding of an input sequence, then *decoded* to the original domain to reconstruct the input sequence. Recent advances in Recurrent Neural Networks (RNNs) [15], especially Long Short-Term Memory (LSTM) [16] and variants [17], have achieved great success in numerous tasks that heavily rely on sentence-representation learning.

RNN-based methods typically model sentences recursively as a generative Markov process with hidden units, where the one-step-ahead word from an input sentence is generated by conditioning on previous words and hidden units, via emission and transition operators modeled as neural networks. In principle, the neural representations of input sequences aim to encapsulate sufficient information about their structure, to subsequently recover the original sentences via decoding. However, due to the recursive nature of the RNN, challenges exist for RNN-based strategies to fully encode a sentence into a vector representation. Typically, during training, the RNN generates words in sequence conditioning on previous *ground-truth* words, *i.e.*, *teacher forcing* training [18], rather than decoding the whole sentence solely from the encoded representation vector. This teacher forcing strategy has proven important because it forces the output sequence of the RNN to stay close to the ground-truth sequence. However, allowing the decoder to access ground truth information when reconstructing the sequence weakens the encoder's ability to produce self-contained representations, that carry enough information to steer the decoder through the decoding process without additional guidance. Aiming to solve this problem, [19] proposed a scheduled sampling approach during training, which gradually shifts from learning via both latent representation and ground-truth signals to solely use the encoded latent representation. Unfortunately, [20] showed that scheduled sampling is a fundamentally inconsistent

training strategy, in that it produces largely unstable results in practice. As a result, training may fail to converge on occasion.

During inference, for which *ground-truth* sentences are not available, words ahead can only be generated by conditioning on previously generated words through the representation vector. Consequently, decoding error compounds proportional to the length of the sequence. This means that generated sentences quickly deviate from the ground-truth once an error has been made, and as the sentence progresses. This phenomenon was coined *exposure bias* in [19].

We propose a simple yet powerful *purely convolutional* framework for learning sentence representations. Conveniently, without RNNs in our framework, issues connected to teacher forcing training and exposure bias are not relevant. The proposed approach uses a Convolutional Neural Network (CNN) [21, 22, 23] as encoder and a deconvolutional (*i.e.*, transposed convolutional) neural network [24, 25] as decoder. To the best of our knowledge, the proposed framework is the first to force the encoded latent representation to capture information from the entire sentence via a multi-layer CNN specification, to achieve high reconstruction quality *without* leveraging RNN-based decoders. Our multi-layer CNN allows representation vectors to abstract information from the entire sentence, irrespective of order or length, making it an appealing choice for tasks involving long sentences or paragraphs. Further, since our framework does not involve recursive encoding or decoding, it can be very efficiently parallelized using convolution-specific Graphical Process Unit (GPU) primitives, yielding significant computational savings compared to RNN-based models.

## 2 Convolutional Auto-encoding for Text Modeling

### 2.1 Convolutional encoder

Let $w^t$ denote the $t$-th word in a given sentence. Each word $w^t$ is embedded into a $k$-dimensional word vector $\boldsymbol{x}_t = \mathbf{W}_e[w^t]$, where $\mathbf{W}_e \in \mathbb{R}^{k \times V}$ is a (learned) word embedding matrix, $V$ is the vocabulary size, and $\mathbf{W}_e[v]$ denotes the $v$-th column of $\mathbf{W}_e$. All columns of $\mathbf{W}_e$ are normalized to have unit $\ell_2$-norm, *i.e.*, $||\mathbf{W}_e[v]||_2 = 1, \forall v$, by dividing each column with its $\ell_2$-norm. After embedding, a sentence of length $T$ (padded where necessary) is represented as $\mathbf{X} \in \mathbb{R}^{k \times T}$, by concatenating its word embeddings, *i.e.*, $\boldsymbol{x}_t$ is the $t$-th column of $\mathbf{X}$.

For sentence encoding, we use a CNN architecture similar to [26], though originally proposed for image data. The CNN consists of $L$ layers ($L - 1$ convolutional, and the $L$th fully-connected) that ultimately summarize an input sentence into a (fixed-length) latent representation vector, $\boldsymbol{h}$. Layer $l \in \{1, \ldots, L\}$ consists of $p_l$ filters, learned from data. For the $i$-th filter in layer 1, a convolutional operation with stride length $r^{(1)}$ applies filter $\mathbf{W}_c^{(i,1)} \in \mathbb{R}^{k \times h}$ to $\mathbf{X}$, where $h$ is the convolution filter size. This yields latent feature map, $\boldsymbol{c}^{(i,1)} = \gamma(\mathbf{X} * \mathbf{W}_c^{(i,1)} + \boldsymbol{b}^{(i,1)}) \in \mathbb{R}^{(T-h)/r^{(1)}+1}$, where $\gamma(\cdot)$ is a nonlinear activation function, $\boldsymbol{b}^{(i,1)} \in \mathbb{R}^{(T-h)/r^{(1)}+1}$, and $*$ denotes the convolutional operator. In our experiments, $\gamma(\cdot)$ is represented by a Rectified Linear Unit (ReLU) [27]. Note that the original embedding dimension, $k$, changes after the first convolutional layer, as $\boldsymbol{c}^{(i,1)} \in \mathbb{R}^{(T-h)/r^{(1)}+1}$, for $i = 1, \ldots, p_1$. Concatenating the results from $p_1$ filters (for layer 1), results in feature map, $\mathbf{C}^{(1)} = [\boldsymbol{c}^{(1,1)} \cdots \boldsymbol{c}^{(p_1,1)}] \in \mathbb{R}^{p_1 \times [(T-h)/r^{(1)}+1]}$.

After this first convolutional layer, we apply the convolution operation to the feature map, $\mathbf{C}^{(1)}$, using the same filter size, $h$, with this repeated in sequence for $L - 1$ layers. Each time, the length along the spatial coordinate is reduced to $T^{(l+1)} = \lfloor (T^{(l)} - h)/r^{(l)} + 1 \rfloor$, where $r^{(l)}$ is the stride length, $T^{(l)}$ is the spatial length, $l$ denotes the $l$-th layer and $\lfloor \cdot \rfloor$ is the floor function. For the final layer, $L$, the feature map $\mathbf{C}^{(L-1)}$ is fed into a fully-connected layer, to produce the latent representation $\boldsymbol{h}$. Implementation-wise, we use a convolutional layer with filter size equals to $T^{(L-1)}$ (regardless of $h$), which is equivalent to a fully-connected layer; this implementation trick has been also utilized in [26]. This last layer summarizes all remaining spatial coordinates, $T^{(L-1)}$, into scalar features that encapsulate sentence sub-structures throughout the entire sentence characterized by filters, $\{\mathbf{W}_c^{(i,l)}\}$ for $i = 1, \ldots, p_l$ and $l = 1, \ldots, L$, where $\mathbf{W}_c^{(i,l)}$ denotes filter $i$ for layer $l$. This also implies that the extracted feature is of fixed-dimensionality, independent of the length of the input sentence.

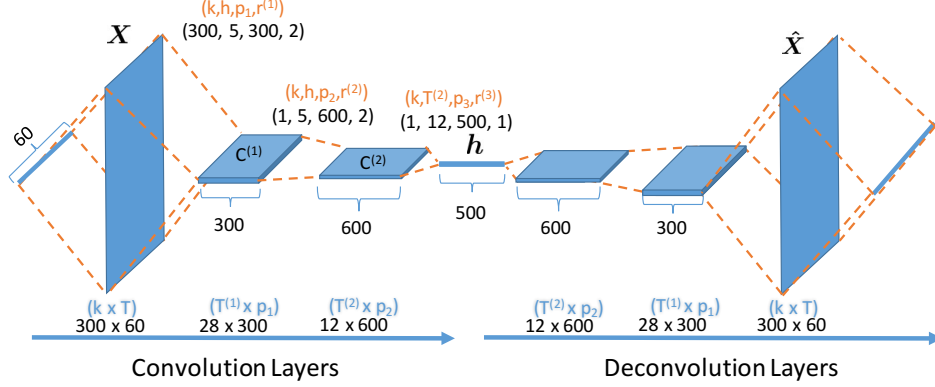

Figure 1: Convolutional auto-encoding architecture. Encoder: the input sequence is first expanded to an embedding matrix, $\mathbf{X}$, then fully compressed to a representation vector $\boldsymbol{h}$, through a multi-layer convolutional encoder with stride. In the last layer, the spatial dimension is collapsed to remove the spatial dependency. Decoder: the latent vector $\boldsymbol{h}$ is fed through a multi-layer deconvolutional decoder with stride to reconstruct $\mathbf{X}$ as $\hat{\mathbf{X}}$, via cosine-similarity cross-entropy loss.

Having $p_L$ filters on the last layer, results in $p_L$-dimensional representation vector, $\boldsymbol{h} = \mathbf{C}^{(L)}$, for the input sentence. For example, in Figure 1, the encoder consists of $L = 3$ layers, which for a sentence of length $T = 60$, embedding dimension $k = 300$, stride lengths $\{r^{(1)}, r^{(2)}, r^{(3)}\} = \{2, 2, 1\}$, filter sizes $h = \{5, 5, 12\}$ and number of filters $\{p_1, p_2, p_3\} = \{300, 600, 500\}$, results in intermediate feature maps, $\mathbf{C}^{(1)}$ and $\mathbf{C}^{(2)}$ of sizes $\{28 \times 300, 12 \times 600\}$, respectively. The last feature map of size $1 \times 500$, corresponds to latent representation vector, $\boldsymbol{h}$.

Conceptually, filters from the lower layers capture primitive sentence information ($h$-grams, analogous to edges in images), while higher level filters capture more sophisticated *linguistic features*, such as semantic and syntactic structures (analogous to image elements). Such a bottom-up architecture models sentences by hierarchically stacking text segments ($h$-grams) as building blocks for representation vector, $\boldsymbol{h}$. This is similar in spirit to modeling linguistic grammar formalisms via concrete syntax trees [28], however, we do not pre-specify a tree structure based on some syntactic structure (*i.e.*, English language), but rather abstract it from data via a multi-layer convolutional network.

## 2.2 Deconvolutional decoder

We apply the deconvolution with stride (*i.e.*, convolutional transpose), as the conjugate operation of convolution, to decode the latent representation, $\boldsymbol{h}$, back to the source (discrete) text domain. As the deconvolution operation proceeds, the spatial resolution gradually increases, by mirroring the convolutional steps described above, as illustrated in Figure 1. The spatial dimension is first expanded to match the spatial dimension of the $(L-1)$-th layer of convolution, then progressively expanded as $T^{(l+1)} = (T^{(l)} - 1) * r^{(l)} + h$, for $l = 1, \cdots$ up to $L$-th deconvolutional layer (which corresponds to the input layer of the convolutional encoder). The output of the $L$-layer deconvolution operation aims to *reconstruct* the word embedding matrix, which we denote as $\hat{\mathbf{X}}$. In line with word embedding matrix $\mathbf{W}_e$, columns of $\hat{\mathbf{X}}$ are normalized to have unit $\ell_2$-norm.

Denoting $\hat{w}^t$ as the $t$-th word in *reconstructed* sentence $\hat{s}$, the probability of $\hat{w}^t$ to be word $v$ is specified as

$$p(\hat{w}^t = v) = \frac{\exp[\tau^{-1} D_{\cos}(\hat{\boldsymbol{x}}^t, \mathbf{W}_e[v])]}{\sum_{v' \in V} \exp[\tau^{-1} D_{\cos}(\hat{\boldsymbol{x}}^t, \mathbf{W}_e[v'])]}, \qquad (1)$$

where $D_{\cos}(\boldsymbol{x}, \boldsymbol{y})$ is the cosine similarity defined as, $\frac{\langle \boldsymbol{x}, \boldsymbol{y} \rangle}{||\boldsymbol{x}|| ||\boldsymbol{y}||}$, $\mathbf{W}_e[v]$ is the $v$-th column of $\mathbf{W}_e$, $\hat{\boldsymbol{x}}^t$ is the $t$-th column of $\hat{\mathbf{X}}$, $\tau$ is a positive number we denote as *temperature* parameter [29]. This parameter is akin to the concentration parameter of a Dirichlet distribution, in that it controls the spread of probability vector $[p(\hat{w}^t = 1) \ \dots \ p(\hat{w}^t = V)]$, thus a large $\tau$ encourages uniformly distributed probabilities, whereas a small $\tau$ encourages sparse, concentrated probability values. In the experiments we set $\tau = 0.01$. Note that in our setting, the cosine similarity can be obtained as an inner product, provided that columns of $\mathbf{W}_e$ and $\hat{\mathbf{X}}$ have unit $\ell_2$-norm by specification. This deconvolutional module can also be leveraged as building block in VAE[30, 31] or GAN[32, 33]

## 2.3 Model learning

The objective of the convolutional autoencoder described above can be written as the word-wise log-likelihood for all sentences $s \in \mathcal{D}$, *i.e.*,

$$\mathcal{L}^{\mathrm{ae}} = \sum_{d \in \mathcal{D}} \sum_t \log p(\hat{w}_d^t = w_d^t), \tag{2}$$

where $\mathcal{D}$ denotes the set of observed sentences. The simple, maximum-likelihood objective in (2) is optimized via stochastic gradient descent. Details of the implementation are provided in the experiments. Note that (2) differs from prior related work in two ways: $i$) [22, 34] use pooling and un-pooling operators, while we use convolution/deconvolution with stride; and $ii$) more importantly, [22, 34] do not use a cosine similarity reconstruction as in (1), but a RNN-based decoder. A further discussion of related work is provided in Section 3. We could use pooling and un-pooling instead of striding (a particular case of deterministic pooling/un-pooling), however, in early experiments (not shown) we did not observe significant performance gains, while convolution/deconvolution operations with stride are considerably more efficient in terms of memory footprint. Compared to a standard LSTM-based RNN sequence autoencoders with roughly the same number of parameters, computations in our case are considerably faster (see experiments) using single NVIDIA TITAN X GPU. This is due to the high parallelization efficiency of CNNs via cuDNN primitives [35].

**Comparison between deconvolutional and RNN Decoders**    The proposed framework can be seen as a complementary building block for natural language modeling. Contrary to the standard LSTM-based decoder, the deconvolutional decoder imposes in general a less strict sequence dependency compared to RNN architectures. Specifically, generating a word from an RNN requires a vector of hidden units that recursively accumulate information from the entire sentence in an order-preserving manner (long-term dependencies are heavily down-weighted), while for a deconvolutional decoder, the generation only depends on a representation vector that encapsulates information from throughout the sentence without a pre-specified *ordering* structure. As a result, for language generation tasks, a RNN decoder will usually generate more coherent text, when compared to a deconvolutional decoder. On the contrary, a deconvolutional decoder is better at accounting for distant dependencies in long sentences, which can be very beneficial in feature extraction for classification and text summarization tasks.

## 2.4 Semi-supervised classification and summarization

Identifying related topics or sentiments, and abstracting (short) summaries from user generated content such as blogs or product reviews, has recently received significant interest [1, 3, 4, 36, 37, 13, 11]. In many practical scenarios, unlabeled data are abundant, however, there are not many practical cases where the potential of such unlabeled data is fully realized. Motivated by this opportunity, here we seek to complement scarcer but more valuable labeled data, to improve the generalization ability of supervised models. By ingesting unlabeled data, the model can learn to abstract latent representations that capture the semantic meaning of all available sentences irrespective of whether or not they are labeled. This can be done prior to the supervised model training, as a two-step process. Recently, RNN-based methods exploiting this idea have been widely utilized and have achieved state-of-the-art performance in many tasks [1, 3, 4, 36, 37]. Alternatively, one can learn the autoencoder and classifier jointly, by specifying a classification model whose input is the latent representation, $\boldsymbol{h}$; see for instance [38, 31].

In the case of product reviews, for example, each review may contain hundreds of words. This poses challenges when training RNN-based sequence encoders, in the sense that the RNN has to abstract information on-the-fly as it moves through the sentence, which often leads to loss of information, particularly in long sentences [39]. Furthermore, the decoding process uses ground-truth information during training, thus the learned representation may not necessarily keep all information from the input text that is necessary for proper reconstruction, summarization or classification.

We consider applying our convolutional autoencoding framework to semi-supervised learning from long-sentences and paragraphs. Instead of pre-training a fully unsupervised model as in [1, 3], we cast the semi-supervised task as a multi-task learning problem similar to [40], *i.e.*, we *simultaneously* train a sequence autoencoder and a supervised model. In principle, by using this joint training strategy, the learned paragraph embedding vector will preserve both reconstruction and classification ability.

Specifically, we consider the following objective:

$$\mathcal{L}^{\text{semi}} = \alpha \sum_{d \in \{\mathcal{D}_l + \mathcal{D}_u\}} \sum_t \log p(\hat{w}_d^t = w_d^t) + \sum_{d \in \mathcal{D}_l} \mathcal{L}^{\text{sup}}(f(\boldsymbol{h}_d), y_d), \qquad (3)$$

where $\alpha > 0$ is an annealing parameter balancing the relative importance of supervised and unsupervised loss; $\mathcal{D}_l$ and $\mathcal{D}_u$ denote the set of labeled and unlabeled data, respectively. The first term in (3) is the sequence autoencoder loss in (2) for the $d$-th sequence. $\mathcal{L}^{\text{sup}}(\cdot)$ is the supervision loss for the $d$-th sequence (labeled only). The classifier function, $f(\cdot)$, that attempts to reconstruct $y_d$ from $\boldsymbol{h}_d$ can be either a Multi-Layer Perceptron (MLP) in classification tasks, or a CNN/RNN in text summarization tasks. For the latter, we are interested in a purely convolutional specification, however, we also consider an RNN for comparison. For classification, we use a standard cross-entropy loss, and for text summarization we use either (2) for the CNN or the standard LSTM loss for the RNN.

In practice, we adopt a scheduled annealing strategy for $\alpha$ as in [41, 42], rather than fixing it *a priori* as in [1]. During training, (3) gradually transits from focusing solely on the unsupervised sequence autoencoder to the supervised task, by annealing $\alpha$ from 1 to a small positive value $\alpha_{\min}$. We set $\alpha_{\min} = 0.01$ in the experiments. The motivation for this annealing strategy is to first focus on abstracting paragraph features, then to selectively refine learned features that are most informative to the supervised task.

## 3   Related Work

Previous work has considered leveraging CNNs as encoders for various natural language processing tasks [22, 34, 21, 43, 44]. Typically, CNN-based encoder architectures apply a single convolution layer followed by a pooling layer, which essentially acts as a detector of specific classes of $h$-grams, given a convolution filter window of size $h$. The deep architecture in our framework will, in principle, enable the high-level layers to capture more sophisticated language features. We use convolutions with stride rather than pooling operators, *e.g.*, max-pooling, for spatial downsampling following [26, 45], where it is argued that fully convolutional architectures are able to learn their own spatial downsampling. Further, [46] uses a 29-layer CNN for text classification. Our CNN encoder is considerably simpler in structure (convolutions with stride and no more than 4 layers) while still achieving good performance.

Language decoders other than RNNs are less well studied. Recently, [47] proposed a hybrid model by coupling a convolutional-deconvolutional network with an RNN, where the RNN acts as decoder and the deconvolutional model as a bridge between the encoder (convolutional network) and decoder. Additionally, [42, 48, 49, 50] considered CNN variants, such as pixelCNN [51], for text generation. Nevertheless, to achieve good empirical results, these methods still require the sentences to be generated sequentially, conditioning on the ground truth historical information, akin to RNN-based decoders, thus still suffering from the exposure bias.

Other efforts have been made to improve embeddings from long paragraphs using unsupervised approaches [2, 52]. The paragraph vector [2] learns a fixed length vector by concatenating it with a *word2vec* [53] embedding of history sequence to predict future words. The hierarchical neural autoencoder [52] builds a hierarchical attentive RNN, then it uses paragraph-level hidden units of that RNN as embedding. Our work differs from these approaches in that we force the sequence to be fully restored from the latent representation, without aid from any history information.

Previous methods have considered leveraging unlabeled data for semi-supervised sequence classification tasks. Typically, RNN-based methods consider either $i)$ training a sequence-to-sequence RNN autoencoder, or a RNN classifier that is robust to adversarial perturbation, as initialization for the encoder in the supervised model [1, 4]; or, $ii)$ learning latent representation via a sequence-to-sequence RNN autoencoder, and then using them as inputs to a classifier that also takes features extracted from a CNN as inputs [3]. For summarization tasks, [54] has considered a semi-supervised approach based on support vector machines, however, so far, research on semi-supervised text summarization using deep models is scarce.

## 4   Experiments

**Experimental setup**    For all the experiments, we use a 3-layer convolutional encoder followed by a 3-layer deconvolutional decoder (recall implementation details for the top layer). Filter size, stride

| Ground-truth: | on every visit to nyc , the hotel beacon is the place we love to stay . so conveniently located to central park , lincoln center and great local restaurants . the rooms are lovely . beds so comfortable , a great little kitchen and new wizz bang coffee maker . the staff are so accommodating and just love walking across the street to the fairway supermarket with every imaginable goodies to eat . |
|---|---|
| Hier. LSTM [52] | every time in new york , lighthouse hotel is our favorite place to stay . very convenient , central park , lincoln center , and great restaurants . the room is wonderful , very comfortable bed , a kitchenette and a large explosion of coffee maker . the staff is so inclusive , just across the street to walk to the supermarket channel love with all kinds of what to eat . |
| Our LSTM-LSTM | on every visit to nyc , the hotel beacon is the place to relax and wanting to become conveniently located . hotel , in the evenings out good budget accommodations . the views are great and we were more than two couples . manny the doorman has a great big guy come and will definitly want to leave during my stay and enjoy a wonderfully relaxing wind break in having for 24 hour early rick's cafe . oh perfect ! easy easy walking distance to everything imaginable groceries . if you may want to watch yours ! |
| Our CNN-DCNN | on every visit to nyc , the hotel beacon is the place we love to stay . so closely located to central park , lincoln center and great local restaurants . biggest rooms are lovely . beds so comfortable , a great little kitchen and new UNK suggestion coffee maker . the staff turned so accommodating and just love walking across the street to former fairway supermarket with every food taxes to eat . |

Table 1: Reconstructed paragraph of the Hotel Reviews example, used in [52].

and word embedding are set to $h = 5$, $r^l = 2$, for $l = 1, \ldots, 3$ and $k = 300$, respectively. The dimension of the latent representation vector varies for each experiment, thus is reported separately.

For notational convenience, we denote our convolutional-deconvolutional autoencoder as CNN-DCNN. In most comparisons, we also considered two standard autoencoders as baselines: $a$) CNN-LSTM: CNN encoder coupled with LSTM decoder; and $b$) LSTM-LSTM: LSTM encoder with LSTM decoder. An LSTM-DCNN configuration is not included because it yields similar performance to CNN-DCNN while being more computationally expensive. The complete experimental setup and baseline details is provided in the Supplementary Material (SM). CNN-DCNN has the least number of parameters. For example, using 500 as the dimension of $h$ results in about 9, 13, 15 million total trainable parameters for CNN-DCNN, CNN-LSTM and LSTM-LSTM, respectively.

| Model | BLEU | ROUGE-1 | ROUGE-2 |
|---|---|---|---|
| LSTM-LSTM [52] | 24.1 | 57.1 | 30.2 |
| Hier. LSTM-LSTM [52] | 26.7 | 59.0 | 33.0 |
| Hier. + att. LSTM-LSTM [52] | 28.5 | 62.4 | 35.5 |
| CNN-LSTM | 18.3 | 56.6 | 28.2 |
| CNN-DCNN | **94.2** | **97.0** | **94.2** |

Table 2: Reconstruction evaluation results on the Hotel Reviews Dataset.

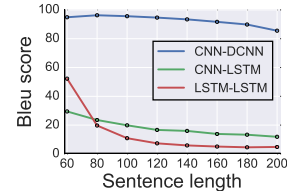

Figure 2: BLEU score *vs.* sentence length for Hotel Review data.

**Paragraph reconstruction** We first investigate the performance of the proposed autoencoder in terms of learning representations that can preserve paragraph information. We adopt evaluation criteria from [52], *i.e.*, ROUGE score [55] and BLEU score [56], to measure the closeness of the reconstructed paragraph (model output) to the input paragraph. Briefly, ROUGE and BLEU scores measures the $n$-gram recall and precision between the model outputs and the (ground-truth) references. We use BLEU-4, ROUGE-1, 2 in our evaluation, in alignment with [52]. In addition to the CNN-LSTM and LSTM-LSTM autoencoder, we also compared with the hierarchical LSTM autoencoder [52]. The comparison is performed on the Hotel Reviews datasets, following the experimental setup from [52], *i.e.*, we only keep reviews with sentence length ranging from 50 to 250 words, resulting in 348,544 training data samples and 39,023 testing data samples. For all comparisons, we set the dimension of the latent representation to $h = 500$.

From Table 1, we see that for long paragraphs, the LSTM decoder in CNN-LSTM and LSTM-LSTM suffers from heavy exposure bias issues. We further evaluate the performance of each model with different paragraph lengths. As shown in Figure 2 and Table 2, on this task CNN-DCNN demonstrates a clear advantage, meanwhile, as the length of the sentence increases, the comparative advantage becomes more substantial. For LSTM-based methods, the quality of the reconstruction deteriorates quickly as sequences get longer. In constrast, the reconstruction quality of CNN-DCNN is stable and consistent regardless of sentence length. Furthermore, the computational cost, evaluated as wall-clock, is significantly lower in CNN-DCNN. Roughly, CNN-LSTM is 3 times slower than CNN-DCNN, and LSTM-LSTM is 5 times slower on a single GPU. Details are reported in the SM.

**Character-level and word-level correction** This task seeks to evaluate whether the deconvolutional decoder can overcome exposure bias, which severely limits LSTM-based decoders. We consider

a *denoising* autoencoder where the input is tweaked slightly with certain modifications, while the model attempts to denoise (correct) the *unknown* modification, thus recover the original sentence.

For character-level correction, we consider the Yahoo! Answer dataset [57]. The dataset description and setup for word-level correction is provided in the SM. We follow the experimental setup in [58] for word-level and character-level spelling correction (see details in the SM). We considered substituting each word/character with a different one at random with probability $\eta$, with $\eta = 0.30$. For character-level analysis, we first map all characters into a 40 dimensional embedding vector, with the network structure for word- and character-level models kept the same.

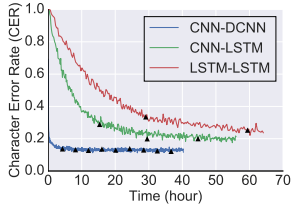

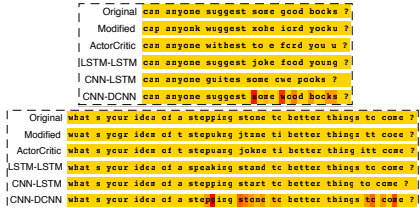

| Model | Yahoo(CER) |
|---|---|
| Actor-critic[58] | 0.2284 |
| LSTM-LSTM | 0.2621 |
| CNN-LSTM | 0.2035 |
| CNN-DCNN | **0.1323** |
| Model | ArXiv(WER) |
| LSTM-LSTM | 0.7250 |
| CNN-LSTM | 0.3819 |
| CNN-DCNN | **0.3067** |

Figure 3: CER comparison. Black triangles indicate the end of an epoch.

Figure 4: Spelling error denoising comparison. Darker colors indicate higher uncertainty. Trained on modified sentences.

Table 3: CER and WER comparison on Yahoo and ArXiv data.

We employ Character Error Rate (CER) [58] and Word Error Rate (WER) [59] for evaluation. The WER/CER measure the ratio of Levenshtein distance (*a.k.a.*, edit distance) between model predictions and the ground-truth, and the total length of sequence. Conceptually, lower WER/CER indicates better performance. We use LSTM-LSTM and CNN-LSTM denoising autoencoders for comparison. The architecture for the word-level baseline models is the same as in the previous experiment. For character-level correction, we set dimension of $h$ to 900. We also compare to actor-critic training [58], following their experimental guidelines (see details in the SM).

As shown in Figure 3 and Table 3, we observed CNN-DCNN achieves both lower CER and faster convergence. Further, CNN-DCNN delivers stable denoising performance irrespective of the noise location within the sentence, as seen in Figure 4. For CNN-DCNN, even when an error is detected but not exactly corrected (darker colors in Figure 4 indicate higher uncertainty), denoising with future words is not effected, while for CNN-LSTM and LSTM-LSTM the error gradually accumulates with longer sequences, as expected.

For word-level correction, we consider word substitutions only, and mixed perturbations from three kinds: substitution, deletion and insertion. Generally, CNN-DCNN outperforms CNN-LSTM and LSTM-LSTM, and is faster. We provide experimental details and comparative results in the SM.

**Semi-supervised sequence classification & summarization** We investigate whether our CNN-DCNN framework can improve upon supervised natural language tasks that leverage features learned from paragraphs. In principle, a good unsupervised feature extractor will improve the generalization ability in a semi-supervised learning setting. We evaluate our approach on three popular natural language tasks: sentiment analysis, paragraph topic prediction and text summarization. The first two tasks are essentially sequence classification, while summarization involves both language comprehension and language generation.

We consider three large-scale document classification datasets: DBPedia, Yahoo! Answers and Yelp Review Polarity [57]. The partition of training, validation and test sets for all datasets follows the settings from [57]. The detailed summary statistics of all datasets are shown in the SM. To demonstrate the advantage of incorporating the reconstruction objective into the training of text classifiers, we further evaluate our model with different amounts of labeled data (0.1%, 0.15%, 0.25%, 1%, 10% and 100%, respectively), and the whole training set as unlabeled data.

For our purely supervised baseline model (supervised CNN), we use the same convolutional encoder architecture described above, with a 500-dimensional latent representation dimension, followed by a MLP classifier with one hidden layer of 300 hidden units. The dropout rate is set to 50%. Word embeddings are initialized at random.

As shown in Table 4, the joint training strategy consistently and significantly outperforms the purely supervised strategy across datasets, even when all labels are available. We hypothesize that during the early phase of training, when reconstruction is emphasized, features from text fragments can be readily

learned. As the training proceeds, the most discriminative text fragment features are selected. Further, the subset of features that are responsible for both reconstruction and discrimination presumably encapsulate longer dependency structure, compared to the features using a purely supervised strategy. Figure 5 demonstrates the behavior of our model in a semi-supervised setting on Yelp Review dataset. The results for Yahoo! Answer and DBpedia are provided in the SM.

| Model | DBpedia | Yelp P. | Yahoo |
|---|---|---|---|
| ngrams TFIDF | 1.31 | 4.56 | 31.49 |
| Large Word ConvNet | 1.72 | 4.89 | 29.06 |
| Small Word ConvNet | 1.85 | 5.54 | 30.02 |
| Large Char ConvNet | 1.73 | 5.89 | 29.55 |
| Small Char ConvNet | 1.98 | 6.53 | 29.84 |
| SA-LSTM (word-level) | 1.40 | - | - |
| Deep ConvNet | 1.29 | 4.28 | 26.57 |
| Ours (Purely supervised) | 1.76 | 4.62 | 27.42 |
| Ours (joint training with CNN-LSTM) | 1.36 | 4.21 | 26.32 |
| Ours (joint training with CNN-DCNN) | **1.17** | **3.96** | **25.82** |

Table 4: Test error rates of document classification (%). Results from other methods were obtained from [57].

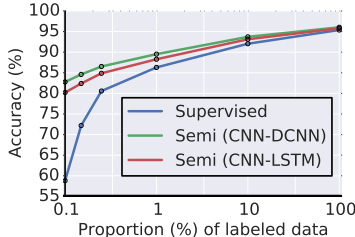

Figure 5: Semi-supervised classification accuracy on Yelp review data.

For summarization, we used a dataset composed of 58,000 abstract-title pairs, from *arXiv*. Abstract-title pairs are selected if the length of the title and abstract do not exceed 50 and 500 words, respectively. We partitioned the training, validation and test sets into 55000, 2000, 1000 pairs each.

We train a sequence-to-sequence model to generate the title given the abstract, using a randomly selected subset of paired data with proportion $\sigma = (5\%, 10\%, 50\%, 100\%)$. For every value of $\sigma$, we considered both purely supervised summarization using just abstract-title pairs, and semi-supervised summarization, by leveraging additional abstracts without titles. We compared LSTM and deconvolutional network as the decoder for generating titles for $\sigma = 100\%$.

Table 5 summarizes quantitative results using ROUGE-L (longest common subsequence) [55]. In general, the additional abstracts without titles improve the generalization ability on the test set. Interestingly, even when $\sigma = 100\%$ (all titles are observed), the joint training objective

| Obs. proportion $\sigma$ | 5% | 10% | 50% | 100% | DCNN dec. |
|---|---|---|---|---|---|
| Supervised | 12.40 | 13.07 | 15.87 | 16.37 | 14.75 |
| Semi-sup. | 16.04 | 16.62 | 17.64 | **18.14** | 16.83 |

Table 5: Summarization task on *arXiv* data, using ROUGE-L metric. First 4 columns are for the LSTM decoder, and the last column is for the deconvolutional decoder (100% observed).

still yields a better performance than using $\mathcal{L}^{sup}$ alone. Presumably, since the joint training objective requires the latent representation to be capable of reconstructing the input paragraph, in addition to generating a title, the learned representation may better capture the entire structure (meaning) of the paragraph. We also empirically observed that titles generated under the joint training objective are more likely to use the words appearing in the corresponding paragraph (*i.e.*, more *extractive*), while the the titles generated using the purely supervised objective $\mathcal{L}^{sup}$, tend to use wording more freely, thus more *abstractive*. One possible explanation is that, for the joint training strategy, since the reconstructed paragraph and title are all generated from latent representation $\boldsymbol{h}$, the text fragments that are used for reconstructing the input paragraph are more likely to be leveraged when "building" the title, thus the title bears more resemblance to the input paragraph.

As expected, the titles produced by a deconvolutional decoder are less coherent than an LSTM decoder. Presumably, since each paragraph can be summarized with multiple plausible titles, the deconvolutional decoder may have trouble when positioning text segments. We provide discussions and titles generated under different setups in the SM. Designing a framework which takes the best of these two worlds, LSTM for generation and CNN for decoding, will be an interesting future direction.

## 5 Conclusion

We proposed a general framework for text modeling using purely convolutional and deconvolutional operations. The proposed method is free of sequential conditional generation, avoiding issues associated with exposure bias and teacher forcing training. Our approach enables the model to fully encapsulate a paragraph into a latent representation vector, which can be decompressed to reconstruct the original input sequence. Empirically, the proposed approach achieved excellent long paragraph reconstruction quality and outperforms existing algorithms on spelling correction, and semi-supervised sequence classification and summarization, with largely reduced computational cost.

**Acknowledgements** This research was supported in part by ARO, DARPA, DOE, NGA and ONR.

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
