[Reviews · NeurIPS 2017]

Reviewer 1



The paper presents an innovative model to endode sentences in vectors that is based on CNNs rather than on RNNs. The proposed model outperforms LSTM in the encoding-decoding task, that is, recostructing sentences from embedding vectors. Yet, there is a major problem: what is the network encoding in the vector? Why has it a better performance in reconstructing the original sentence? How does it encode the notion of sequence? There are many NN models and many NN papers. Yet, the reasons why these models are better are obscure. Encoding sentences in vectors has been also a realm of compositional distributional semantics. Moreover, there are works that attempt to encode syntactic structures in vectors and recover them back [1]. This is done without learning. [1]Ferrone et al. (2015) Decoding Distributed Tree Structures. In: Dediu AH., Martín-Vide C., Vicsi K. (eds) Statistical Language and Speech Processing. Lecture Notes in Computer Science, vol 9449. Springer, Cham

Reviewer 2



This paper proposes a purely Convolutional-Deconvolutional framework for encoding/decoding sentences. The main contributions of this paper are to propose the deconvolution module as a decoder of sentences and to show the effectiveness of this deconvolution module. In particular, when the output sentence length is long, the deconvolution module performs significantly better than RNN-based decoders do. RNN-based decoder works poorly when the output sentence length is long, which is known as the exposure bias. Deconvolution decoders do not rely on neighborhood predicted for producing output sentences; therefore, preidction mistakes do not propagate the subsequent outputs. This advantage enables to avoid the exposure bias. As a result, this characteristic leads to the effective semi-supervised learning framework for leveraging plenty of unlabeled text resources to solve various NLP tasks, even when these resources contain long sentences. This paper is well-written. The discussion is well-organized. The contributions of this paper are summarized clearly. The effectiveness of the proposed method is carefully validated by well-designed experiments. I think the quality of this paper is clearly above the threshold of the acceptance. I have one minor concern: does the deconvolutional module outperform the RNN with attention, too? Although the exposure bias occurs even when we use the attention mechanism, the attention might boost the performance of RNN-based seq2seq methods. If the authors could add the comparison between the deconvolutional module and the RNN with attention, it would be better.

Reviewer 3



This paper proposes encoder-decoder models that uses convolutional neural networks, not RNNs. Experimental results on NLP data sets show that the proposed models outperform several existing methods. In NLP, RNNs are more commonly used than CNNs. This is because the length of an input varies. CNNs can only handle a fixed length of an input, by truncating input text or padding shortage of the input. In experiments, it is okay to assume the length of an input, such as 50-250 words; however, in reality, we need to handle an arbitrary length of text information, sometimes 10 and sometimes 1,000 words. In this respect, RNNs, especially, LSTMs, have high potential in NLP applications. Major Comments: 1. For summarization experiments, Document Understanding Conference (DUC) data sets and CNN news corpus are commonly used. 2. For the headline summarization, if you use shared data, we can judge how your models are better than other methods, including non-neural approaches. 3. Table 2 shows that the CNN-DCNN model can directly output an input. Extremely high BLEU scores imply low generalization. It has a sufficient rephrasing ability, it cannot achieve the BLUE scores over 90. 4. In image processing, similar models have already been proposed, such as: V. Badrinarayanan, A. Handa, and R. Cipolla. SegNet: a deep convolutional encoder-decoder architecture for robust semantic pixel-wise labelling. arXiv preprint arXiv:1505.07293, 2015. Please make your contributions clear in your paper. 5. I am not sure how your semi-supervised experiments have been conducted. In what sense, your methods are semi-supervised? 6. Regarding the compared LSTM-LSTM encoder-decoder model, is it an attention-based encoder-decoder model? 7. I suggest to try machine translation data sets, in your future work.